# Occurrence of *Listeria* spp. and *Listeria monocytogenes* Isolated from PDO Taleggio Production Plants

**DOI:** 10.3390/foods9111636

**Published:** 2020-11-10

**Authors:** Erica Tirloni, Cristian Bernardi, Francesco Pomilio, Marina Torresi, Enrico P. L. De Santis, Christian Scarano, Simone Stella

**Affiliations:** 1Department of Health, Animal Science and Food Safety, Università degli Studi di Milano, Via Celoria 10, 20133 Milan, Italy; cristian.bernardi@unimi.it (C.B.); simone.stella@unimi.it (S.S.); 2Istituto Zooprofilattico Sperimentale dell’Abruzzo e del Molise “G. Caporale”, Via Campo Boario, 64100 Teramo, Italy; f.pomilio@izs.it (F.P.); m.torresi@izs.it (M.T.); 3Department of Veterinary Medicine, University of Sassari, Via Vienna 2, 07100 Sassari, Italy; desantis@uniss.it (E.P.L.D.S.); scarano@uniss.it (C.S.)

**Keywords:** traceability, *Listeria monocytogenes*, *Listeria* spp., whole genome sequencing (WGS), PDO Taleggio cheese

## Abstract

The present study evaluated the presence of *Listeria* spp. and *L. monocytogenes* in four plants producing PDO Taleggio cheese. A total of 360 environmental samples were collected from different areas during production. The sampling points were identified as Food Contact Surfaces (FCS), transfer-Non Food Contact Surfaces (tr-NFCS), and non-transfer-NFCS (non-tr-NFCS). Fifty-nine ingredients/products were also analyzed. *Listeria* spp. was found in all the plants with a mean prevalence of 23.1%; plants that included a ripening area showed significantly higher prevalence if compared to the other plants. The positivity rate detected on FCS was moderate (~12%), but significantly lower if compared to NFCS (about 1/4 of the samples, *p* < 0.01). Among the FCS, higher prevalence was revealed on ripening equipment. *Listeria* spp. was never detected in the ingredients or products. A total of 125 *Listeria* spp. isolates were identified, mostly as *L. innocua* (almost 80%). *L. monocytogenes* was detected only from two FCS samples, in an area dedicated to the cutting of ripened blue cheeses; strain characterization by whole genome sequencing (WGS) evidenced a low virulence of the isolates. The results of the present study stress the importance of *Listeria* spp. management in the dairy plants producing PDO Taleggio and similar cheeses, mainly by the application of strict hygienic practices.

## 1. Introduction

*Listeria monocytogenes* remains a food safety challenge for cheese production plants: dairy products contaminated by this pathogen have been associated to some strong-evidence listeriosis outbreaks that occurred in Europe in the last decade [1,2]. In 2018, data from 16,486 sampling units of soft and semi-soft cheeses (including fresh cheese) were supplied to European Food Safety Authority (EFSA), highlighting the occurrence of *L. monocytogenes* in soft and semi-soft cheeses made with raw or mild heat-treated milk (0.8%) and with pasteurized milk (0.3%) [2]. Little et al. [3] stated that 2% of dairy products produced from raw or thermized milk were characterized by *L. monocytogenes* counts above the limits reported by EU Regulation 2073/2005 [4]; in addition, dairy products produced from pasteurized milk should be considered as potential sources of outbreaks, as reported by previous authors [5,6]. In 2017, Amato et al. published a retrospective study that revealed a major listeriosis outbreak in Northern Italy between 2009 and 2011 was linked to soft cheese, which was undetected by local authorities: based on epidemiological outcomes, the authors reasonably hypothesized that PDO Taleggio cheese was the implicated food [7].

Thanks to its ability to replicate at refrigeration conditions and tolerate high salt concentrations, low water activity (below 0.92), and a wide range of pH values (4.0–9.5), it is not surprising that *L. monocytogenes* may survive and persist in dairy processing environments, or even grow when they find a favorable substrate, as in several dairy products [8,9].

Dairy products contamination can be considered a multifactorial event. Many sources are involved: raw milk can be contaminated at farm level and lead the contamination in the cheese-processing environment [10]; this event could have a negative impact mainly on the production of dairy products from raw or unpasteurized milk. The contamination by *L. monocytogenes* may also occur at several stages of dairy production and can be due to the use of contaminated ingredients or products (such as starter cultures or brines), but it is mainly caused by the spread from contaminated environment, utensils, and equipment such as floors, drains, shelves, clothes, brushes, coolers, etc. [11,12]. Previous studies reported prevalence up to 52% in dairy plants [12,13,14,15].

Focusing on the production environment, several studies have already shown the presence of *L. monocytogenes* in cheese production plants during processing but also after cleaning and disinfection procedures [16,17,18,19], due to the ability of the pathogen to adhere to various food contact surfaces and to produce biofilm. Furthermore, the previous exposure to sublethal conditions (high salt concentration, low temperatures, etc.) that are typical of dairy plants may act as a protection against additional stresses such as heat, pH, acidic, and oxidative stresses [10], thanks to the ability of the pathogen to perform a metabolic adaptation [20]. Many studies have indicated the persistence of *L. monocytogenes* strains up to 10 years in the food industries [11,12].

According to EU Regulation 2073/2005 [4], the number of *L. monocytogenes* should not exceed 100 CFU/g during the shelf-life of RTE foods such as dairy products. In addition, *L. monocytogenes* should not be detectable in the product samples (*n* = 5.25 g each), if the foodstuff is able to support its growth, when it is still under the control of the producer, who cannot also demonstrate, to the satisfaction of the competent authority, that the counts will not exceed the limit of 100 CFU/g throughout the shelf-life. In order to apply an efficient control of the pathogen, processing areas and equipment should be monitored in addition to cheese testing; as other *Listeria* species (in particular *L. innocua*) share the same ecological niches with *L. monocytogenes*, it is recommended to enlarge the research in dairy plants to *Listeria* spp. to investigate on the presence of ecological niches suitable for the survival of the pathogen [21].

The aim of the present study was to detect the presence of *Listeria* spp. and *L. monocytogenes* within the processing environment of four different plants involved in the production chain of PDO Taleggio cheese: three producers and one ripening plant. Taleggio is a PDO cheese produced in a limited area of Northern Italy, but is marketed worldwide. For the production, calf rennet and bacterial starters (*Lactobacillus bulgaricus* and *Streptococcus thermophilus*) are added to whole cow milk (raw or mainly pasteurized). The curd obtained is distributed into molds and submitted to resting (at 22–25 °C, for 8–16 h) and to dry- or brine-salting. Then, the cheese is ripened for a minimum of 35 days on wood axes or boxes at refrigeration temperatures; during this time, it is weekly turned and sponged with brine [22]. The production and ripening phases expose this product to potential surface contamination by *Listeria monocytogenes*. This contamination could be of concern, especially considering that Taleggio rind is intended for consumption; thus, tracing the contamination within the plants could contribute to the hygienic management of the production process.

## 2. Materials and Methods

### 2.1. Samples Collection

Samples were collected from four different dairy plants during three sampling sessions occurred in May, July, and October 2017. Briefly, dairy plant “A” produces PDO Taleggio, other PDO cheeses (Salva Cremasco and Quartirolo Lombardo), and non-PDO blue cheese; it is intended for both production and ripening. Dairy plant “B” produces various PDO cheeses (Taleggio, Salva Cremasco, Quartirolo Lombardo and Gorgonzola), other cheeses that need to be further ripened, and ricotta: ripening is done in plant “D.” Dairy plant “C” only produces PDO Taleggio cheese and a small amount of whey-butter; ripening occurs in plant “D.” Plant D only has premises for ripening and packaging different cheeses, such as PDO Taleggio, PDO Gorgonzola, PDO Strachitunt, and others, supplied by several producers (not only “B” and “D”).

Environmental samples were collected during the dairy production process: a total of 360 environmental samples were collected from different areas (milk pasteurization, curd/cheese production, brining, ripening, packaging, equipment cleaning). The sampling points were identified as Food Contact Surfaces (FCS), transfer-Non Food Contact Surfaces (tr-NFCS, points that can favor the transfer of the contamination among different areas, such as boots, trolleys, etc.), and non-transfer-NFCS (non-tr-NFCS) (such as walls, floors, drains, etc.). In particular, considering the production areas, sampling was performed on tables, drains, instruments, boxes, filters, manhole covers, washing machines, floors, scales, and operators’ boots. Samplings from the ripening areas involved boxes, brushes, packaging machines, sheets, and operators’ boots. Moreover, when present, other rooms/areas not directly involved in Taleggio production were sampled (in these areas, PDO Gorgonzola, ricotta, or whey butter were produced). In addition to environmental samples, a total of 59 ingredients (11 pasteurized milk, 12 salt or brine), curd (10) and product samples (22 Taleggio rinds at the end of the ripening period and 4 Ricotta) were taken.

### 2.2. Sample Analysis

For the detection of *Listeria* spp. and *L. monocytogenes*, sterile sponges pre-wetted with 10 mL of Buffer Peptone Water (BPW) (Biogenetics, Ponte San Nicolò, Italy) were used to swab surfaces, when possible, of about 10 × 10 cm^2^. After sampling, the sponges were immediately put into the original sterile bag and kept refrigerated. For the enumeration of *Listeria* spp. and *L. monocytogenes*, the same points (near the ones used for the detection) were sampled with pre-wet sponges following the same procedure. All the samples were immediately transported in refrigeration at the laboratory and analyzed within 24 h of collection.

For bacterial detection samples were analyzed according to ISO 11290-1:2017 [23]. After incubation, an aliquot of the broth (pre-enrichment and selective enrichment) was streaked onto Palcam Agar (Biogenetics) and ALOA (Biogenetics) plates, for the detection of *Listeria* spp. and *L. monocytogenes*, respectively. For enumeration purposes, aliquots of 90 mL of BPW were added; appropriate 10-fold dilutions were prepared and spread onto Palcam Agar (for *Listeria* spp.) and on ALOA (for *L. monocytogenes*). From each positive sample, at least one to three presumptive *Listeria* spp. and *L. monocytogenes* colonies were taken and streaked onto Triptic Soy Agar (TSA, Scharlab, Sentmenat, E) and incubated at 37 °C for 24 h. Each colony was confirmed after gram staining, catalase and oxidase test, hemolytic activity and biochemical test Microgen Listeria-ID (MID67) (Microbiogen products, Novacyt group, Microbiogen products, Novacyt group, Surrey, UK).

### 2.3. L. monocytogenes Serogroup and Whole Genome Sequencing (WGS)

Strains identified as *L. monocytogenes* by biochemical methods were subjected to DNA extraction, molecular serogrouping and WGS to define in silico Clonal Complex (CC), Sequence Type (ST), cgMLST, and relevant genes absence/presence, as previously described by Torresi et al. [24]. Briefly, strains were grown overnight in sheep blood agar (Microbiol & C., Cagliari, I), picked and dissolved in 300 µL of nuclease-free water (Ambion^TM^, Thermo Fisher Scientific, Waltham, MA, USA). Then, 100 µL of 20 mg/mL lysozyme was added and incubated for 2 h at 56 °C. Finally, 300 µL of the suspension were transferred to the cartridges provided by the Maxwell 16 Cell DNA Purification Kit (Promega, Madison, WI, USA). DNA extraction was accomplished following the manufacturer’s instructions. DNA was quantified using a Qubit dsDNA HS (High Sensitivity) Assay Kit (Invitrogen, Carlsbad, CA, USA) and purity was checked by a Nanodrop ND-1000 spectrophotometer (Thermo Fisher Scientific).

Molecular serogroup-was performed using a multiplex PCR assay according to Doumith et al. and Kerouanton et al. [25,26]. Briefly, DNA was extracted as described before and reactions were carried out in a Gene Amp PCR System 9700 thermal cycler (Applied Biosystems, Foster City, CA, USA). PCR products were run in 2% (w/v) agarose gel in 1X TBE buffer (Biorad, Hercules, CA, USA) and visualized by SYBR^TM^ Safe DNA Gel Stain (Thermo Fisher Scientific).

WGS was performed to define in silico Clonal Complex (CC), Sequence Type (ST), cgMLST and relevant genes absence/presence.

Strains DNA were sequenced using the NextSeq500 Illumina platform using the Nextera XT protocol. Quality control was performed by FastQC [27] and trimming was carried out by Trimmomatic [28]. A de novo assembly was performed by Spades 3.11 [29] and in silico MLST 7 loci was defined. The cgMLST analysis was carried out by chewBBACA [30] using the Pasteur Institute scheme for *L. monocytogenes*. Then, genomes were uploaded into the Pasteur Institute platform (https://bigsdb.pasteur.fr) [31] in order to define presence/absence of virulence-, metal-, and detergent-resistance genes and stress islands.

### 2.4. Statistical Analysis

Data were submitted to chi-square test or exact Fisher’s test using PRISM graph pad 6 (GraphPad Software, San Diego, CA, USA); the prevalence data obtained from the different plants, during the three sampling sessions, and the different areas within each plant were compared. A comparison was also made among FCS, tr-NFCS and non-tr-NFCS samples. The threshold for statistically significant differences was settled at *p* < 0.05.

## 3. Results

### 3.1. Prevalence of Listeria spp. in the Dairy Plants

The results obtained from the detection of *Listeria* spp. and *L. monocytogenes* in the samples are summarized in Table 1 and Table 2. *Listeria* spp. was found in all the plants in at least one sampling session (with a prevalence ranging from 0 to 73.1%), with a mean prevalence of 23.1% (Table 1), thus indicating its wide diffusion in this plants. The prevalence of positive samples for *Listeria* spp. was significantly higher in plants A and D, if compared to plants B and C (*p* < 0.05). This result was strongly influenced by the presence, in plants A and D, of the cheese ripening area. *Listeria monocytogenes* was found in only two samples from plant D (6.9%).

When considering the same areas in the different dairy plants, no significant differences were shown related to the production (plants A-B-C) or ripening (plants A–D) areas: these results confirm the important influence of the production or ripening phase rather than the producer considered, taking into account that all the plants apply currently good manufacturing and sanitation practices. Focusing on the ripening phase, the entry of cheeses from different plants did not represent a particular concern, as the prevalence in plant D, which receives unripened Taleggio from various plants (B, C and others), was lower (if not significantly) than that observed in plant A, whose ripening area was dedicated exclusively to the cheese produced in the same plant.

The sampling session showed a significant effect on *Listeria* spp. prevalence (Table 1). In particular, the mean prevalence in October was significantly higher (*p* < 0.05) in the plants including a ripening area (A and D), whereas this influence was not evidenced in the plants dedicated only to the production phase (B and C). When considering the areas separately, a significant difference was evidenced only in the samples from the ripening one, with data obtained from samplings performed in October showing higher counts than those taken in July and in May (*p* < 0.05). Thus, this finding could not be related to a different initial contamination rate of the product; as ripening environmental conditions are strictly maintained, the possible role of management procedures during this phase should be further investigated.

Considering the different areas dedicated to the phases of PDO Taleggio production within each plant (Table 2), the data confirmed the major importance of the ripening areas, with a significantly higher mean prevalence (*Listeria* spp. was detected in more than half of the samples) if compared to pasteurization, brining, packaging, and production areas. This could be explained by the high risk of contamination of the product in the ripening rooms from the environment, which could also be increased by cross-contaminations when commingled products from other plants are introduced. The ripening rooms are very often used for the aging of various typologies of cheeses that share the same needs for environmental parameters; thus, the cross contamination among different products is not unlikely. The environmental conditions of the ripening rooms, such as low temperatures and high ambient humidity, coupled with the surface type, and their contact time with the product may enhance the possibility for *L. monocytogenes* to persist and be selected. In this area frequent handling of the product (e.g., rind brushing) also occurred, which can contribute to the transfer of pathogens, with equipment or devices (brushes) that are difficult to sanitize and where the structures cannot be submitted to daily sanitation (e.g., shelves).

As expected, lower positivity rates were revealed in the production and in the milk pasteurization areas, thanks to the absence of ripened product and to the efficacy of the daily cleaning and disinfection procedures, thus limiting the presence of suitable ecological niches.

### 3.2. Prevalence and Counts of Listeria spp. on FCS and NFCS

The data obtained from the analysis were categorized considering the classification of the sampling points as Food Contact Surfaces (FCS), transfer-Non Food Contact Surfaces (tr-NFCS), and non-transfer-NFCS (non-tr-NFCS). As shown in Figure 1, the positivity rate detected on FCS was moderate (about 12%), but significantly lower than that detected on NFCS (about 1/4 of the samples, *p* < 0.01), as expected. Considering the different areas of the plants, the lower rate on FCS was mainly evidenced in the ripening areas (*p* < 0.05); the comparison among the three sampling sessions showed that the higher prevalence observed during the last session (performed in October, see Section 3.1) was mainly linked to the non-tr-NFCS, thus suggesting the role of environmental niches within the plants.

Among the FCS, a very low positivity rate was recorded on equipment/surfaces located in the production and brining areas, whereas a high prevalence was detected on equipment used for ripening like brushes, tables, boxes, boards, and cloths (8/26 positive samples) (Table 2). In a previous study by De Cesare et al. [32], 52 environmental swabs collected in a PDO Taleggio plant from salting equipment, ripening cloths, and ripening boxes were tested for the presence of *L. monocytogenes* showing 80% of positivity; in particular, they selected 16 strains and traced a contamination of the salting equipment. A particularly high prevalence of *Listeria* spp. was detected in equipment used for Taleggio rind brushing (6/8 positive samples on brushes and brushing tables). These data are in agreement with those obtained with other dairy products submitted to surface washing (positivity rate of 44–72%) [16,17,18,19], highlighting the importance of such equipment to monitor the potential contamination of the product. The brushing procedure of Taleggio is made periodically (every 10–15 days) during ripening, and it is traditionally performed manually. Proper practices, such as frequent change/disinfection of brushes, gloves, and working surfaces, must be currently applied by the producer in this phase, aiming to limit the cross contamination among the different cheese batches. An encouraging result was obtained from the sampling of operators’ gloves and hands, as no *Listeria* spp. was detected in any of the areas of the plants (0/14 samples), thus indicating the compliance with the personnel hygiene recommendations.

Considering the results obtained from the non-tr-NFCS surfaces, a low prevalence of *Listeria* spp. was detected for some typologies (tables, ducts, wall), with only 3/37 positive samples (Table 2). An interesting finding was obtained from the analysis of the outer surface of water hoses, as 2 out of 13 samples showed the presence of the target microorganism: this result could be justified by the fact that the final part of the hose comes often in contact with the floor. It has to be considered that water hoses are used during each working session and in the pauses, and are then frequently handled by the operators, thus representing a potential source of contamination. The highest prevalence rates of *Listeria* spp. among the non-tr-NFCS were detected, as expected, in the samples taken from the floors (9/31) and especially from the drains (36/109); in applying the procedures of the HACCP plan, the choice of the drains as monitoring points should be regarded as an effective method to reveal the presence of *Listeria* within a dairy plant. With the same aim, the sampling of the overshoes used during the sampling activity, reflecting the situation of the floors of all the production, ripening, and packaging areas, demonstrated to be an even more sensitive method, resulting in a very high prevalence of *Listeria* spp. (11/18 samples).

A particular concern could be posed by tr-NFCS surfaces, as they can act as by-pass points for the contamination of different working areas: the highest prevalence among these samples was detected on trolley wheels (4 out of 16 samples), followed by operators’ boots (3/17) and doors/handles (1/7) (Table 2). It has to be noted that trolleys are not cleaned as frequently as walls, floors, and doors and handles, and are used to carry equipment, cheese boxes, packaging materials, etc. in all the areas of the plants (often including the external parts during the trucks loading and unloading procedures). Thus, they represent potential carriers for bacterial entry and spread within the production process.

The presence of various production processes into the same plant should be carefully considered when evaluating the contamination patterns: in this study, three areas dedicated to the production of different dairy products were considered: the contamination of ricotta and butter production rooms (in plant B and C, respectively) showed a very low contamination rate (just one positive sample, taken from the drain in the ricotta production area, out of 20). This result was expected, as both butter and ricotta production processes are characterized by a low contamination probability if considered alone, but in particular ricotta must be protected by the post process contamination coming from other working areas, as it represents a suitable growth substrate for *L. monocytogenes* [33].

The room dedicated to the piercing, cutting, and packaging of blue cheeses (mainly PDO Gorgonzola) showed relatively high *Listeria* spp. contamination rates, both from FCS (3/9) and drains (3/5) (Table 2); in addition, *L. monocytogenes* was detected from the two FCS samples, namely the piercing machine and the table where the cheese is placed during cutting. This finding must be taken into account when evaluating the risk for both blue cheeses (as this equipment is used for several cheese wheels before being sanitized) and other products obtained in the same plant (due to the possible transfer of the microorganism). In this study, despite the proximity to the Taleggio packaging area, a spread of *L. monocytogenes* from this section was not observed, suggesting the importance of good production practices in segregating the contamination within complex plants.

Finally, *Listeria* spp. was never detected in the ingredients (salt, saline, milk) or in the product, both considering the curd and Taleggio; these results confirm the importance of the application of proper procedures to manage the presence of these microorganisms in the production plants (as it cannot be completely avoided in the current process conditions) and to prevent the risk for consumers.

In most of the positive samples, *Listeria* spp. counts were below the enumeration limit of the method (1 CFU/cm^2^), except for some sampling points (17 out of 83 positive samples, less than 5% of the total number of environmental samplings), where counts between 1 and 1000 CFU/cm^2^ were recorded. These samples were obtained only from plants A and D (18.8% and 33.3%, respectively, of the total positive samples), confirming the difference already noticed in the bacterial prevalence. High counts were mainly obtained from the ripening areas, characterized by environmental conditions supporting the selection and slow growth of *Listeria* spp., and on floor drains, where microorganisms are easily collected during the cleaning procedures. In drains, counts varied from 1 (drains in packaging) to 700 CFU/cm^2^ (drains in blue cheese piercing machine). Three samples with high *Listeria* spp. counts were taken from FCS, and in particular from brushes (20 and 240 CFU/cm^2^) and a table where cheese blocks were laid during brushing operations (86 CFU/cm^2^), confirming the need for particular care during this periodical production phase of PDO Taleggio. In blue cheese, piercing machine values of 110 CFU/cm^2^ were enumerated, while in sampler overshoes, a value of 1 CFU/cm^2^ was enumerated.

### 3.3. Identification of Listeria spp. Isolates

A total of 125 *Listeria* spp. isolates were submitted to species identification (Table 3): they were mostly identified as *Listeria innocua*, which represented almost 80% of the isolates, and was found in all the plants during quite all the sampling sessions in which *Listeria* spp. was detected. The other isolates were identified as *Listeria ivanovii* (mainly from a single plant during one sampling session), *L. grayi*, *L. welshimeri*, *L. seeligeri*, and *L. monocytogenes*. The diffusion of the different *Listeria* species among the sampling points was also evaluated. As shown in Figure 2, *L. innocua* was predominant in all the areas of the plants (63 out of 83 positive samples hosted *L. innocua*); apart from the abovementioned presence of *L. ivanovii*, the two most contaminated areas showed the presence of a diversified *Listeria* spp. population, namely the ripening area and the one dedicated to piercing, cutting, and packaging of blue cheeses. These data confirm the importance of cheese ripening for the selection and persistence of *Listeria*, thanks to the environmental conditions and to the possibility to survive steadily on the surface of the equipment and potentially on the cheese blocks/wheels.

In any case, as stated previously, the presence of *Listeria* spp. may act as a sentinel for the presence of the pathogenic strains as they share the same ecological niches. Moreover, although *Listeria innocua* was initially considered non-pathogenic, its role has been debated, as it was isolated at least once from a fatal human case [34].

### 3.4. Listeria monocytogenes Characterization

As shown in Table 3, *L. monocytogenes* strains accounted for 1.6% (*n* = 2) of the overall isolates (*n* = 125). Both strains were isolated from the blue cheese production area and were characterized as belonging to serogroup IIa. Molecular characterization “in silico” revealed genetic proximity of the two isolates. Both strains were classified as CC31, ST325 and showed the same virulence, antibiotic resistance, and stress resistance genes. In particular, among 92 loci classified in the Pasteur Institute platform as belonging to virulence, 51 were detected, comprising a complete *Listeria* Pathogenicity Island 1 (LIPI 1) and InlB, E, F, H, J, K. InlA was detected but showed the same single point mutation in both strains. As regard the stress islands, four of the five genes belonging to SSI 1 (lmo0444-lmo04448) were detected as exact match while gaps or two single mutations were recorded for the gene lmo0445. No metal and detergent resistance genes were detected. Core genome MLST analysis showed three allele differences due to the lack of three loci in one of the isolates. Characterization results highlighted the finding of hypovirulent strains, a 2017 EFSA opinion [35,36] showed as CC31 *L. monocytogenes* strains were rarely found in humans and are usually present in meat and meat products. CC31 strains were, furthermore, found among non-hemolytic strains [37]. As specified before, no BC efflux pumps qac (Tn6188), bcrABC, and emrE were found. On the contrary, the presence of four genes of SSI 1 implies adaptation to low pH and high salt concentration, even if the lack of an exact match for the gene lmo0445 of SSI 1 encoding a positive regulator of the islet genes could affect the capacity to effectively respond to adverse environmental conditions.

## 4. Conclusions

The results of the present study show the diffusion of *Listeria* spp. in different PDO Taleggio producing plants, with very low prevalence of *L. monocytogenes*. The data analysis showed the importance of the ripening areas, of which the role was quite independent from other factors, such as the plant or the origin of the curd. A moderate prevalence of *Listeria* spp. on the Food Contact Surfaces: in this light, it has to be considered that the contamination of the cheese mass by *L. monocytogenes* is unlikely; however, the rind contamination (e.g., during brushing) can in any case be considered of concern, as PDO Taleggio rind is defined to be edible, and a link between this cheese and listeriosis outbreaks has been hypothesized in some cases. The isolation of *L. monocytogenes* strains from the plant environment, even if hypo-virulent, confirms the importance of the application of control procedures along the production and ripening process. Moreover, the identification of clones previously isolated from a totally different food production chain puts the focus on the ability of this pathogen to adapt to various hurdles, thus colonizing different environments.

The identification of the potential sources of *L. monocytogenes* in a dairy processing plant is crucial for the application of corrective and preventive measures and for the traceability of outbreak clones. Moreover, the overlay of the production areas dedicated to different products is likely present in several dairy plants, making possible the cross contamination of the final products. Three of the plants considered in this study produced several different cheeses other than PDO Taleggio; such products are potentially affected by the same risk factors for the growth of *Listeria monocytogenes* (environmental contamination sources and a long ripening phase at mild refrigeration). In some cases, such products can act as additional niches for *Listeria* persistence within the plants or as vehicles for the contamination of PDO Taleggio itself. In this light, particular attention must be posed to some transfer surfaces, which can favor the circulation of the microorganism among areas dedicated to different dairy products and production phases. Thus, a careful cleaning and disinfection program, as well as the proper management of personnel, equipment, and product flows, must be applied. A constant microbiological monitoring program in the plant should be also addressed as one of the main control points in a dairy plant. In such program, *Listeria* spp. can act as a sensitive sentinel for the identification of potential survival niches, that could be hardly identified by the only research of *L. monocytogenes*.

The results of the present study, performed in medium scale plants that currently follow specific hygiene procedures, stress the importance of *Listeria* spp. management in the dairy plants producing PDO Taleggio and similar cheeses, mainly by the application of strict hygienic practices (involving both sanitation and production procedures), the implementation of operator training, and the management of personnel, equipment, and product flows, in order to guarantee product safety.

## Figures and Tables

**Figure 1 foods-09-01636-f001:**
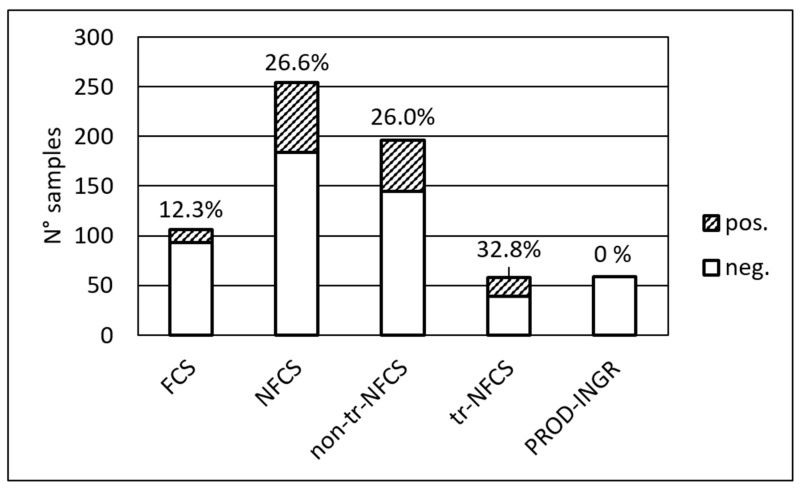
*Listeria* spp. detection rate in FCS, NFCS, and product/ingredients samples. Food Contact Surfaces (FCS), Non Food Contact Surfaces (NFCS), transfer-Non Food Contact Surfaces (tr-NFCS), and non-transfer-NFCS (non-tr-NFCS), product ingredients (PROD-INGR).

**Figure 2 foods-09-01636-f002:**
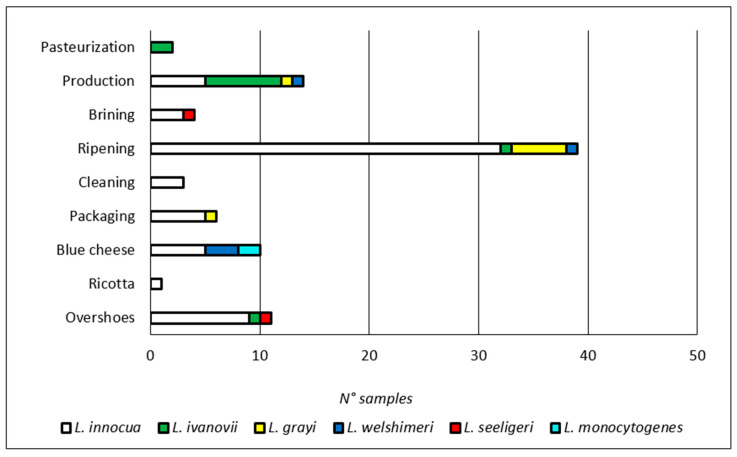
Distribution of the different *Listeria* species within the plant areas.

**Table 1 foods-09-01636-t001:** Prevalences of *Listeria* spp. and *L. monocytogenes* in the dairy plants in the three sampling sessions.

Plant	Month	Number of Samples	*Listeria* spp.		*Listeria monocytogenes*	
			Positive Samples	%	Positive Samples	%
**A**	May	40	4	10.0 ^b^	0	0
	July	34	9	26.5 ^b^	0	0
	October	26	19	73.1 ^a^	0	0
	All A	100	32	32.0 ^y^	0	0
**B**	May	38	2	5.3	0	0
	July	32	2	6.3	0	0
	October	26	5	19.2	0	0
	All B	96	9	9.4 ^z^	0	0
**C**	May	31	8	25.8 ^a^	0	0
	July	25	0	0 ^b^	0	0
	October	24	1	4.2	0	0
	All C	80	9	11.3 ^z^	0	0
**D**	May	29	6	20.1 ^b^	0	0
	July	29	11	37.9	2	6.9
	October	26	16	61.5 ^a^	0	0
	All D	84	33	39.3 ^y^	2	2.4
**Total**	May	138	20	14.5 ^b^	0	0
	July	120	22	18.3 ^b^	2	6.9
	October	102	41	40.2 ^a^	0	0
	All	360	83	23.1	2	0.6

Statistically significant differences among the sampling sessions are indicated by superscripts ^a,b^ (*p* < 0.05); statistically significant differences among the producing plants are indicated by superscript ^y,z^ (*p* < 0.05).

**Table 2 foods-09-01636-t002:** Positivity rate for *Listeria* spp. in the different areas and sample typologies of the dairy plants and number of isolates.

Area	Typology	Sample	N° Positive for *Listeria* spp./ Analyzed Samples					N° of Isolates
			Plant A	Plant B	Plant C	Plant D	Total	
**Milk pasteurization**	FCS	Milk filter	-	-	0/1	-	0/1	-
	Non-tr-NFCS	Water hose	-	-	1/2	-	1/2	1
		Drain	-	0/6	1/4	-	1/10	1
	Total		-	0/6	2/7	-	2/13 (15.4%) ^b^	**2**
**Production**	FCS	Curdling tank	0/3	0/1	0/1	-	0/5	-
		Table-grind-box-board	1/7	0/4	0/8	-	1/19	1
		Whey collection duct (internal surface)	0/2	0/7	0/5	-	0/14	-
		Operator’s hand	0/2	0/3	0/2	-	0/7	-
	Tr-NFCS	Door-handle	0/1	0/2	0/2	-	0/5	-
		Trolley wheels	1/4	0/4	1/5	-	2/13	2
		Operator’s boots	0/2	0/4	0/3	-	0/9	-
	Non-tr-NFCS	Table-boxes (external surface)	0/4	0/4	0/2	-	0/10	-
		Whey collection duct (external surface)	0/3	1/6	1/5	-	2/14	2
		Water hose	0/3	0/3	0/2	-	0/8	-
		Wall (drip)	0/3	0/1	1/5	-	1/9	1
		Floor	0/5	1/8	1/2	-	2/15	2
		Drains	1/7	3/16	1/19	-	5/42	10
	Total		3/46	5/63	5/61	-	13/170 (7.6%) ^b,c^	**18**
**Brining**	FCS	Brining vat (internal surface)	0/3	0/3	-	-	0/6	-
	Tr-NFCS	Plastic doors	1/2	-	-	-	1/2	1
	Non-tr-NFCS	Brining vat (external surface)	1/1	0/1	-	-	1/2	1
		Drains	2/4	-	0/4	-	2/8	5
	Total		4/10	0/4	0/4	-	4/18 (22.2%) ^b^	7
**Ripening**	FCS	Brushing table	3/3	-	-	-	3/3	6
		Brush	3/3	-	-	0/2	3/5	5
		Operator’s gloves	0/2	-	-	0/3	0/5	-
		Plastic boxes-board-cheese cloth	2/7	-	-	0/6	2/13	2
	Tr-NFCS	Operator’s boots	1/2	-	-	2/3	3/5	4
	Non-tr-NFCS	Water hose	1/2	-	-	-	1/2	1
		Floor	5/9	-	-	2/6	7/15	7
		Drains	3/4	-	-	15/24	18/28	41
	Total		18/32	-	-	19/44	37/76 (48.7%) ^a^	**66**
**Cleaning**	Non-tr-NFCS	Equipment washing machine	-	0/1	-	0/3	0/4	-
		Drains	-	1/1	-	2/3	3/4	**3**
	Total		-	1/2	-	2/6	3/8 (37.5%) ^b^	3
**Packaging**	FCS	Table/conveyor	-	-	-	1/6	1/6	1
		Cheese cutter	-	-	-	0/2	0/2	-
		Operator’s hands	-	-	-	0/2	0/2	-
	Tr-NFCS	Trolley wheels	2/3	-	-	-	2/3	2
		Operator’s boots	-	-	-	0/1	0/1	-
	Non-tr-NFCS	Drain	1/3	-	-	2/6	3/9	3
	Total		3/6	-	-	3/17	6/23 (26.1%) ^b^	**6**
**Blue cheese piercing, cutting and packaging**	FCS	Table	-	-	-	1/3	1/3	1(1)
		Cheese piercing-cutting equipment	-	-	-	1/5	1/5	1(1)
		Board	-	-	-	1/1	1/1	2
	Non-tr-NFCS	Drains	-	-	-	3/5	3/5	6
	Total		-	-	-	6-2 */14	6-2 */14 (42.9%) ^a^	**10-2 ***
**Ricotta/butter production**	FCS	Butter churn-ricotta smoothing equipment	-	0/4	0/3	-	0/7	-
	FCS/non-tr-NFCS	Handle-Operator’s hand	-	0/1	0/1	-	0/2	-
	Tr-NFCS	Operator’s boots	-	0/2	-	-	0/2	-
	Non-tr-NFCS	Table-cooler tank	-	0/4	-	-	0/4	-
		Water hose	-	0/1	-	-	0/1	-
		Floor	-	0/1	-	-	0/1	-
		Drains	-	1/3	-	-	1/3	1
	Total	-	-	1/16	0/4	-	1/20 (5%) ^b,c^	**1**
**Other**	Tr-NFCS	Sampler overshoes	4/6	2/5	2/4	3/3	11/18 (61.1%) ^a^	**12**
**Ingredients**	Salt		0/1	0/1	-	-	0/2	-
	Brine (brining)		-	0/3	0/2	-	0/5	-
	Brine (ripening)		0/2	-	-	0/3	0/5	-
	Milk		0/1	0/6	0/4	-	0/11	-
	Total		0/4	0/10	0/6	0/3	0/23	-
**Product**	Ricotta		-	0/4	-	-	0/4	-
	Curd		0/3	0/4	0/3	-	0/10	-
	Final product (PDO Taleggio)		0/5	0/1	0/4	0/12	0/22	-

* Isolates identified as *Listeria monocytogenes*. Statistically significant differences among the plant areas (excluding the ones intended for the production of other cheeses/dairy products) are indicated by superscripts ^a,b,c^ (*p* < 0.05).

**Table 3 foods-09-01636-t003:** Identification of *Listeria* spp. isolates.

Plant	Species	May	July	October	General
A	*L. innocua*	4/4 (100%)	12/14 (85.7%)	24/31 (77.4%)	40/49 (81.6%)
	*L. ivanovii*	-	-	4/31 (12.9%)	4/49 (8.2%)
	*L. grayi*	-	-	2/31 (6.5%)	2/49 (4.1%)
	*L. welshimeri*	-	-	1/31 (3.2%)	1/49 (2.0%)
	*L. seeligeri*	-	2/14 (14.3%)	-	2/49 (4.1%)
B	*L. innocua*	-	2/2 (100%)	4/5 (80%)	6/9 (66.7%)
	*L. grayi*	1/2 (50%)	-	-	1/9 (11.1%)
	*L. welshimeri*	-	-	1/5 (20%)	1/9 (11.1%)
	*L. seeligeri*	1/2 (50%)	-	-	1/9 (11.1%)
C	*L. innocua*	-	-	1/1 (100%)	1/9 (11.1%)
	*L. ivanovii*	8/8 (100%)	-	-	8/9 (88.9%)
D	*L. innocua*	13/13 (100%)	8/15 (53.3%)	29/30 (96.7%)	50/58 (86.2%)
	*L. grayi*	-	4/15 (26.7%)	-	4/58 (6.9%)
	*L. welshimeri*	-	1/15 (6.7%)	1/30 (3.3%)	2/58 (3.45%)
	*L. monocytogenes*	-	2/15 (13.3%)	-	2/58 (3.45%)
Total	*L. innocua*	17/27 (63.0%)	22/31 (71.0%)	58/67 (86.5%)	97/125 (77.6%)
	*L. ivanovii*	8/27 (29.6%)	-	4/67 (6.0%)	12/125 (9.6%)
	*L. grayi*	1/27 (3.7%)	4/31 (12.9%)	2/67 (3.0%)	7/125 (5.6%)
	*L. welshimeri*	-	1/31 (3.2%)	3/67 (4.5%)	4/125 (3.2%)
	*L. seeligeri*	1/27 (3.7%)	2/31 (6.45%)	-	3/125 (2.4%)
	*L. monocytogenes*	-	2/31 (6.45%)	-	2/125 (1.6%)

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
