# Peer review of "Occurrence of Listeria spp. and Listeria monocytogenes Isolated from PDO Taleggio Production Plants"

_foods, 2020, doi:10.3390/foods9111636_

Round 1

Reviewer 1 Report

The article focuses on the issue of occurrence of Listeria spp. and Listeria monocytogenes isolated from PDO Taleggio production plants. Authors focused in particular on the monitoring of listeria in swab samples from different parts of production plants, taken both from places coming into direct contact with foods and from other places that can serve as a source of cross-contamination.

I have a few comments on the work:

  • page 3, chapter 2.2.: the detection of Listeria monocytogenes was carried out in accordance with ISO 11290-1:2017, it is not clear from the text whether the subsequent inoculation was performed not only from Half Fraser Broth (primary step) or subsequently also from Fraer Broth (second step)
  • page 3, chapter 2.2.: when determining the number, a spread was performed on PALCAM and ALOA agar, how much inoculum was used (1 ml according to Regulation 2073/2005?)
  • page 3, chapter 2.3.: "lysozyme was added and incubated for 2 h at 56 C" - correctly 56 °C
  • page 3, chapter 2.3.: it is common to use a special character "1×TBE"
  • table 1: in line Plane A is a redundant line
  • table 2: in line Milk pasteurization as for the other lines, the results for FCS are not separated (the given line is short)
  • page 12, chapter 3.3.: "A total of 125 Listeria spp. ...." reference to Table 4 only
  • figure 2: poorly graphically represented, differences between individual species of listeria are not well recognizable

Author Response

  • page 3, chapter 2.2.: the detection of Listeria monocytogenes was carried out in accordance with ISO 11290-1:2017, it is not clear from the text whether the subsequent inoculation was performed not only from Half Fraser Broth (primary step) or subsequently also from Fraer Broth (second step)

        we clarifid the protocol procedure

  • page 3, chapter 2.2.: when determining the number, a spread was performed on PALCAM and ALOA agar, how much inoculum was used (1 ml according to Regulation 2073/2005?)
    yes

  • page 3, chapter 2.3.: "lysozyme was added and incubated for 2 h at 56 C" - correctly 56 °C

        this was corrected

  • page 3, chapter 2.3.: it is common to use a special character "1×TBE"

        this was corrected

  • table 1: in line Plane A is a redundant line

       this was corrected

  • table 2: in line Milk pasteurization as for the other lines, the results for FCS are not separated (the given line is short)

       this was corrected

  • page 12, chapter 3.3.: "A total of 125 Listeria spp. ...." reference to Table 4 only

        this was done

  • figure 2: poorly graphically represented, differences between individual species of listeria are not well recognizable

        Figure was improved

Reviewer 2 Report

The work presented in the manuscript “Occurrence of Listeria spp. and Listeria monocytogenes isolated from PDO Taleggio production plants” is really relevant, although it is not entirely innovative. In general, the text can be improved, as sometimes it is a little confusing and difficult to follow. I suggest some minor changes for improvement:

Last paragraph of introduction– “The aim of…” instead of “Aim of…”

MM, Section 2.2., Line 11 – “L. monocytogenes” instead of “monocytogenes”

MM, Section 2.3.:

Line 6 – “56 °C” instead of “56 C”.

Line 11 – “Doumith et al. Kerouanton et al.” or “Doumith et al. and Kerouanton et al.”?

Line 16 – “WGS was performed to define in silico Clonal Complex (CC), Sequence Type (ST), cgMLST and relevant genes absence/presence.” This sentence is repeated (line 2). Authors must delete one of the sentences.

Results, Section 3.1.:

Why do the authors do not discuss the results for L. monocytogenes? Please change it accordingly.

Line 3 – “with a mean prevalence of 23.1%” instead of “with a mean prevalence of 23.1”.

Table 2 – Italicize L. monocytogenes in the caption at the end of the table.

Author Response

Last paragraph of introduction– “The aim of…” instead of “Aim of…”

This was done

MM, Section 2.2., Line 11 – “L. monocytogenes” instead of “monocytogenes”

This was done

MM, Section 2.3.:

Line 6 – “56 °C” instead of “56 C”.

This was done

Line 11 – “Doumith et al. Kerouanton et al.” or “Doumith et al. and Kerouanton et al.”?

This was done

Line 16 – “WGS was performed to define in silico Clonal Complex (CC), Sequence Type (ST), cgMLST and relevant genes absence/presence.” This sentence is repeated (line 2). Authors must delete one of the sentences.

This was done

Results, Section 3.1.:

Why do the authors do not discuss the results for L. monocytogenes? Please change it accordingly.

We included a sentence in that section

Line 3 – “with a mean prevalence of 23.1%” instead of “with a mean prevalence of 23.1”.

This was done

Table 2 – Italicize L. monocytogenes in the caption at the end of the table.

This was done

Reviewer 3 Report

The focus of the manuscript with Number FOODS 980055, related to the occurrence of Listeria spp. and Listeria monocytogenes in cheese production and ripening plants is very interesting.

In general, the manuscript is very well written. However, the following considerations should be taken into account:

  1. Title: Could the authors justify why PDO Taleggio appear in the title since the samples evaluated were taken in 4 different plants where: two are for different types of cheese production and one also for different types of cheese ripening, while only in one of those four plants (named C) PDO Taleggio cheese is exclusively produced?  

2. In the Introduction section:

            -Line 1: (L.monocytogenes): should remove since it is not necessary there.

3. M and M section:

 section 2.2. second paragraph, line 4: there is the lack of L. for L.monocytogenes

  section 2.3.      - line 3: et al. in cursive

                        -line 6: 56ºC

4. In Results and Discussion section: In general, more recent reference should be included/changed)

- Could the authors present an explanation for the much higher prevalence of Listeria spp in the samples taken in October? And why it was in plant C in May?

- Could the authors justify that the prevalence of Listeria spp. in the samples was because of the PDO Taleggio cheese production? The prevalence was lower in plant C than in plants A and D.

Author Response

  1. Title: Could the authors justify why PDO Taleggio appear in the title since the samples evaluated were taken in 4 different plants where: two are for different types of cheese production and one also for different types of cheese ripening, while only in one of those four plants (named C) PDO Taleggio cheese is exclusively produced?  

The inclusion of taleggio in the title is due to the fact is the main prodction of each of the dairy plants studied, as well as the only product produced by all the plants  considered 

Moreover the possiible cross contamination among different products was faced in the discussion.

2. In the Introduction section:

            -Line 1: (L.monocytogenes): should remove since it is not necessary there. done

3. M and M section:

 section 2.2. second paragraph, line 4: there is the lack of L. for L.monocytogenes this was included

  section 2.3.      - line 3: et al. in cursive done

                        -line 6: 56ºC done

4. In Results and Discussion section: In general, more recent reference should be included/changed)

  • Could the authors present an explanation for the much higher prevalence of Listeria spp in the samples taken in October? And why it was in plant C in May this was included 
  •  
  • Could the authors justify that the prevalence of Listeria spp. in the samples was because of the PDO Taleggio cheese production? we included a sentence in the conclusion to clarify the potential role of other cheeses.

Reviewer 4 Report

The article submitted to Foods entitled “Occurrence of Listeria spp. and Listeria monocytogenes isolated from PDO Taleggio production plants” is an interesting article adding information to a problem that is well known in the dairy industry: Listeria's persistence in the plants and its potential spreading to cheese, with severe consequences for the consumer.

The article is well organized, studying an adequate number of sampling points in four plants. The identification of Listeria species was performed, as well the identification of virulence factors.

English is not my natural language. Having that into consideration, I consider the article is very well written.

Just some small suggestions:

#1 If I understood correctly the plant D is an “affinage” unit (excuse-me the French word…). It could be included in the introduction before it appears in MM section.

#2 In the last part of the text in the introduction, please consider to explain the technology of Tallegio first, and the aim of the article at the end. This segment could be used to that note on the “affinage” units. Please consider indicating that it is made from cow milk (as I believe it is??)

#3 In MM section 2.1. some examples of tr-NFCS and non-tr-NFCS could improve the understanding of the experiment (or a table with the exact objects/surfaces were included in each category. Only when the reader look at the results understands exactly what kind of samples were taken.

#4 Table 1. Are the data on table 1 a synthesis of those presented in table 2 in a detailed form; The comparison between the month of sampling has marginal importance. It was used as a “repetition” in this study. In my opinion, the use of capital letters and lowercase letters to differentiate 0.01 from 0.05 do not add very important information to the results, and it complicates the reading of the article, once the reader is expecting that upper and lower case letters are comparing different thinks. Please consider presenting all as p<0.05 (that is the limit defined in the statistical analysis in the MM section).

Once a comparison between months was done, a justification for these observations could be pointed out. Is it related to the relative humidity? or to the milking stage?... please consider speculating the why of these differences. If authors believe that it was random/accidental, the statistical comparison is not necessary

#5. Table 2 is quite long. I believe that the information would be easier to understand if the data are presented for the type of samples (FCS, tr-NFCS, and non-tr-NFCS). For me, as a reader, it is not important id the strain of Listeria was detected on a hose or in a wall!; Please consider shortening this table merging the values of each sampling category.

#6 still in table 2. The heading Listeria s.. (L. monocytogenes)/analyzed samples.. Is not easy to understand. Only when the reader checks the footnote on the table understands that L. monocytogenes is only in the indications with *; I only found one asterisk in “Blue cheese piercing cutting and packaging. Please consider to delete L- monocytogenes from the heading, once with a small distraction, the reader might consider that all the cases reported are the pathogenic species of listeria.

#7 Figure 1 and 2. Please consider using the color of a contrasting pattern. In the present form, it is basically impossible to read.

#8. The value of the information in table 3 is limited. It could be easily replaced by two or three-line lines in the text giving that note (if the authors consider really necessary); Most of the situations with counts are in drains, that are reasonably expected.

Author Response

#1 If I understood correctly the plant D is an “affinage” unit (excuse-me the French word…). It could be included in the introduction before it appears in MM section. yes we included in introduction this aspect

#2 In the last part of the text in the introduction, please consider to explain the technology of Tallegio first, and the aim of the article at the end. This segment could be used to that note on the “affinage” units. Please consider indicating that it is made from cow milk (as I believe it is??) this was done (lines 77-81)

#3 In MM section 2.1. some examples of tr-NFCS and non-tr-NFCS could improve the understanding of the experiment (or a table with the exact objects/surfaces were included in each category. Only when the reader look at the results understands exactly what kind of samples were taken. examples were included

#4 Table 1. Are the data on table 1 a synthesis of those presented in table 2 in a detailed form; The comparison between the month of sampling has marginal importance. It was used as a “repetition” in this study. In my opinion, the use of capital letters and lowercase letters to differentiate 0.01 from 0.05 do not add very important information to the results, and it complicates the reading of the article, once the reader is expecting that upper and lower case letters are comparing different thinks. Please consider presenting all as p<0.05 (that is the limit defined in the statistical analysis in the MM section).Once a comparison between months was done, a justification for these observations could be pointed out. Is it related to the relative humidity? or to the milking stage?... please consider speculating the why of these differences. If authors believe that it was random/accidental, the statistical comparison is not necessary. We think these information may be useful especially for readers that would like to compare own data of a specific season with our data: for this reason we think it could be interesting to maintain these data expressed for month; we presented all as p<0.05                                                                            5. Table 2 is quite long. I believe that the information would be easier to understand if the data are presented for the type of samples (FCS, tr-NFCS, and non-tr-NFCS). For me, as a reader, it is not important id the strain of Listeria was detected on a hose or in a wall!; Please consider shortening this table merging the values of each sampling category.6 still in table 2. The heading Listeria s.. (L. monocytogenes)/analyzed samples.. Is not easy to understand. Only when the reader checks the footnote on the table understands that L. monocytogenes is only in the indications with *; I only found one asterisk in “Blue cheese piercing cutting and packaging. Please consider to delete L- monocytogenes from the heading, once with a small distraction, the reader might consider that all the cases reported are the pathogenic species of listeria.

Table 2: we agree for the heading

for the data presented: we tried to reduce but we think we would loose information that could be particurlly useful for the reader

#7 Figure 1 and 2. Please consider using the color of a contrasting pattern. In the present form, it is basically impossible to read. We tried to improve the figure to make it easier to understand

#8. The value of the information in table 3 is limited. It could be easily replaced by two or three-line lines in the text giving that note (if the authors consider really necessary); Most of the situations with counts are in drains, that are reasonably expected. We deleted table 3 and we included some lines in the text
